# Global Analysis of the WOX Transcription Factor Gene Family in *Populus × xiaohei* T. S. Hwang et Liang Reveals Their Stress−Responsive Patterns

**Yue Li, Chunhui Jin, Yuting Liu, Lili Wang, Fangrui Li, Bo Wang, Guifeng Liu, Jing Jiang and Huiyu Li ***

State Key Laboratory of Tree Genetics and Breeding, Northeast Forestry University, 51 Hexing Road, Harbin 150040, China; qq2845833005@163.com (Y.L.); bluesky_jin7@126.com (C.J.); Liu5774565699@163.com (Y.L.); wdnssyz@126.com (L.W.); kylinli1997@163.com (F.L.); wb19940314@126.com (B.W.); liuguifeng@126.com (G.L.); jiangjing1960@126.com (J.J.)
* Correspondence: lihuiyu2020@nefu.edu.cn

**Abstract:** The WUSCHEL−related homeobox (WOX) family is a group of plant−specific transcription factors that play important regulatory roles in embryo formation, stem cell stability, and organogenesis. To date, there are few studies on the molecular mechanisms involved in this family of genes in response to stress. Thus, in this study, eight WOX genes were obtained from an endemic Chinese resilient tree species, *Populus × xiaohei* T. S. Hwang et Liang. Bioinformatic analysis showed that the WOX genes all contained a conserved structural domain consisting of 60 amino acids, with some differences in physicochemical properties. Phylogenetic analysis revealed that WOX members were divided into three evolutionary clades, with four, one, and three members in the ancient, intermediate, and modern evolutionary clades, respectively. The conserved structural domain species as well as the organization and gene structure of WOX genes within the same subfamily were highly uniform. Chromosomal distribution and genome synteny analyses revealed seven segmental−duplicated gene pairs among the *PsnWOX* gene family that were mainly under purifying selection conditions. Semi−quantitative interpretation (SQ−PCR) analysis showed that the WOX gene was differentially expressed in different tissues, and it was hypothesized that the functions performed by different members were diverse. The family members were strongly and differentially expressed under $CdCl_2$, NaCl, $NaHCO_3$, and PEG treatments, suggesting that WOX genes function in various aspects of abiotic stress defense responses. These results provide a theoretical basis for investigating the morphogenetic effects and abiotic stress responses of this gene family in woody plants.

**Keywords:** *Populus × xiaohei* T. S. Hwang et Liang; WOX family; gene clone; abiotic stress; expression analysis

## 1. Introduction

The WUSCHEL−related homeobox (WOX) family of transcription factors belongs to the Homeobox (HOX) superfamily. The typical homeodomain of the HOX superfamily has 60 amino acid residues that fold into a "helix−loop−helix−turn−helix" spatial structure, where a combination of the second and third helix forms a "helix−turn−helix" that can bind to specific DNA sequences [1–5]. WUSCHEL (WUS) is the most primitive gene in the WOX transcription factor family. In 2004, Haecker used a homology search method to identify 15 other members with similar structures using the WUS gene in the *Arabidopsis thaliana* genome and named them WOX genes [6]. As research continued, 31, 10, and 29 WOX genes were identified in *Zea mays* [7], *Lycopersicon esculentum,* and *Camellia sinensis*, respectively. Based on phylogenetic analysis, the WOX protein family was divided into three clades: the ancient, intermediate, and modern clades [8]. The distribution of WOX proteins contained in the three branches of plant taxonomic groups differed significantly, with all plants containing teleost WOX proteins. However, *Chlorella* and *Physcomitrella patens* contained

only the ancient clade WOX proteins, and the intermediate clade proteins were found in higher ferns and seed plants, while the WUS clade WOX proteins were specific to higher seed plants [9,10].

As the function of WOX family genes continues to be studied, it has been demonstrated that they play a range of crucial roles in the development and growth of plants, such as in embryonic development patterns, organ formation, and stem cell maintenance [11–14]. *WOX1* plays a role in the growth and regulation of leaf width in *A. thaliana* [15], *Pharbitis nil*, and *Nicotiana tabacum*. *WOX4* is expressed in almost all of the vascular systems of the plant, and it has been found to have specific expression sites in the cambium and procambium tissues, i.e., sites that affect secondary plant growth [16,17]. *WOX5* is mainly expressed in roots and maintains stem cell stability in the apical meristematic tissue. It is also expressed during the growth and development of the lateral root primordium and cotyledon primordium, indicating its involvement in the development of these tissues [18–20]. *WOX9* is not only involved with *WOX8* in regulating the development of *A. thaliana* from fertilized eggs to embryos, but is also expressed in the root tip meristem tissue and can promote root cell multiplication [21,22]. *WOX13* is involved in the development of lateral roots and flowering organ formation in *A. thaliana*, and the results of Deveaux's study confirm that the *WOX13* orthologous group is the most conserved WOX clade [23,24].

WOX transcription factors broadly regulate postembryonic developmental processes in plants and enable the plant body to adapt to environmental changes under abiotic stresses through the corresponding signal transduction pathways [1]. The functional studies of WOX genes in abiotic stresses are still in their infancy. In 2004, Zhu discovered that the *AtWOX6* allele mutant strain was sensitive to freezing, revealing its involvement in low−temperature response; this initiated the investigation of WOX in abiotic stress responses [6]. *WOX11* regulates the growth and development of rice crown root by repressing the expression of RR2 gene, and its overexpression increases the resistance to drought stress [25,26]. In a study of tomato, drought induced the expression of *SlWOX13* gene [27]. To date, there are even fewer studies on WOX genes in abiotic stress response in woody plants. Overexpression of the *JrWOX11* gene in *Juglans regia* prompted advancing adventitious root primordia formation by two days, and also improved resistance to osmotic stress and salt tolerance in 84K poplar. The expression of *PtoWOX11/12a* gene in roots and leaves was significantly increased in *Populus tomentosa* under salt and drought treatments, and its overexpression transgenic poplar SOD and POD enzyme activities as well as proline were higher than the control, indicating that *PtoWOX11/12a* induced expression under salt and drought [28].

Based on the genomic data in the public platform, genome−wide analyses such as target gene member identification, gene structure, and expression patterns were performed to lay the foundation for revealing their related functions and regulatory mechanisms. In the latest genome−wide study, Kumar et al. identified the PIN gene family in wheat and found that the function of the PIN family is essential in regulating the distribution of polar growth hormone [29]. Based on genome−wide analysis, You et al. found that the *MeNRT2* gene family may play a role in nitrate uptake and utilization, and based on this, low nitrogen treatment of trans−*MeNRT2.2* strains showed that *A. thaliana* overexpressing the *MeNRT2.2* gene exhibited higher fresh weight compared to the wild type, further corroborating the findings obtained from genome−wide [30]. The results of Feng et al. provided new insights into the role played by GIR gene members in cotton fiber formation and provided ideas for future research using breeding tools for cotton fiber improvement [31].

*Populus × xiaohei* was produced from a cross between *Populus simonii* and *Populus nigra*. The hybrid tree has many desirable characteristics, including rapid growth, drought resistance, and resistance to pests and diseases. It is an excellent fast−growing green species in the sandy and arid regions of northern China and is also an ideal material for studying the response mechanisms of woody plants to abiotic stresses. To date, the sequencing of the genome of *Populus × xiaohei* has not been completed; thus, in the present study, eight WOX genes were obtained by cloning with reference to the information of the genome of *Populus*

*trichocarpa* and were analyzed by bioinformatic methods. In addition, we also determined the expression characteristics of WOX genes in different tissue sites and under various abiotic stresses. The results provide a theoretical reference for functional studies of WOX genes in woody plant growth and development as well as in abiotic stress responses of *Populus* × *xiaohei*.

## 2. Materials and Methods

### 2.1. Cloning, Identification, and Physicochemical Characterization of PsnWOX Sequences

Based on the fact that the genome of *Populus* × *xiaohei* has not been released, we referred to the sequence of *P. trichocarpa* for identification. To identify WOX proteins from *P. trichocarpa*, the protein database of *P. trichocarpa* was downloaded from Phytozome v12.0 (https://phytozome.jgi.doe.gov/pz/portal.html (accessed on 11 January 2022)) [32], and the Hidden Markov Model files were downloaded from the Pfam database (https://pfam.xfam.org/ (accessed on 8 January 2022)) [33]. Using the hmmsearch command in HMMER v3.1 software [34], the poplar protein database was searched, and the identified sequences were integrated.

Specific primers were designed for the 8 identified *P. trichocarpa* sequences (Table S1). The full−length Open Reading Frame sequences of WOX genes were amplified using cD-NAs from different tissues of *Populus* × *xiaohei* as templates, using KOD−Plus high−fidelity enzyme (KOD−201, TOYOBO (SHANGHAI) BIOTECH CO. LTD., Shanghai, China). The reaction system consisted of 10 × PCR Buffer for KOD−Plus− 1.7 µL, template 0.5 µL, 25 mM MgSO$_4$ 0.8 µL, 2 mM dNTPs 1.7 µL, KOD−Plus− 0.4 µL, 13.7 µL double distilled water, and 0.6 µL each of upstream and downstream primers. Reaction conditions were as follows: 94 °C pre−denaturation for 2 min, followed by 35 cycles of 94 °C denaturation for 15 s, 58 °C annealing for 30 s and 68 °C extension for 1 Kb/1 min. The target bands were then constructed into the pMD−18T vector, and the accuracy of the sequences was verified by Polymerase Chain Reaction (PCR) and sequencing.

Tools from the ExPasy website (https://www.expasy.org/ (accessed on 5 November 2021)) were used to obtain the sequence lengths, molecular weights, and isoelectric points of the identified WOX proteins [35]. The online tool WOLF PSORT (https://www.genscript.com/psort/wolf_psort.html (accessed on 5 November 2021)) was used to predict subcellular localization [36].

### 2.2. Multiple Sequence Alignment and Phylogenetic Analysis of PsnWOX Proteins

Multiple sequence alignment of *PsnWOX* was performed using BioEdit software; secondary structure was generated through the PRABI (https://npsa-prabi.ibcp.fr/cgi-bin/npsa_automat.pl?page=npsa_sopma.html (accessed on 10 November 2021)) online website; sequence logos was generated using the Weblogo (http://weblogo.berkeley.edu/logo.cgi (accessed on 10 November 2021)) online website.

To reveal the evolutionary relationships of WOX genes between *Populus* × *xiaohei* and other species, a Neighbor−Joining (NJ) tree was constructed using MEGA v7.0.14 with the number of bootstrap replicates set to 1000 [37].

### 2.3. Exon/Intron Structure and Conserved Motifs Analysis

The online program GSDS 2.0 (http://gsds.cbi.pku.edu.cn (accessed on 12 November 2021)) was used to predict the distribution patterns of exons and introns in the *PsnWOX* genes [38]. The intron insertion information of the WOX genes of *A. thaliana* and *Populus* × *xiaohei* was obtained from Phytozome and the reference for *P. trichocarpa*, respectively. The online software MEME v5.0.5 (http://meme-suite.org/tools/meme (accessed on 12 November 2021)) was used to identify conserved motifs [39].

### 2.4. Chromosomal Location, Duplication Analysis, and Ka/Ks Calculation

Location information for the *PsnWOX* genes was retrieved from Phytozome and PopGenIE (http://popgenie.org/chromosome-diagram (accessed on 10 January 2022))

with reference to the *P. trichocarpa* [40]. The MG2C tool (http://mg2c.iask.in/mg2c_v2.0 (accessed on 10 January 2022)) was used to construct a chromosome distribution map of the *PsnWOX* genes [41]. Chromosome gff3 files of *P. trichocarpa* and other species were obtained from EnsemblPlants (https://plants.ensembl.org (accessed on 10 January 2022)). Then, gene replication events were analyzed using the Multiple Collinear Scanning toolkit (http://chibba.pgml.uga.edu/mcscan2/ (accessed on 10 January 2022)) under the Linux system. The Ka and the Ks values were calculated using DnaSP [42].

### 2.5. Plant Materials, Growth Conditions, and Stress Treatment

*Populus × xiaohei* plants were grown at the Northeast Forestry University Genetics and Breeding Laboratory, Harbin, China. The current year shoots were cut into spikes containing 2–3 axillary buds, and the cuttings were inserted into 5 cm × 5 cm nutrient bowls containing the same weight of substrate (black soil: humus: vermiculite = 2:1:1). After 2 months of incubation in a greenhouse with 16 h of light/8 h of darkness at 25 °C, plants of uniform growth and free from diseases and pests were selected as materials for gene cloning, tissue extraction, and abiotic stress treatment.

A total of five tissues were selected from the terminal bud, the third leaf, xylem, phloem, and root of *Populus × xiaohei*, and three sets of samples were collected from each tissue as three biological replicates, which were snap−frozen in liquid nitrogen and stored in −80 °C refrigerator for studying the expression characteristics of WOX gene in different tissue parts. Stem segments and leaves from the 1st–11th nodes and 1st–14th leaves of the whole plant were selected to simulate the whole developmental process from young to senescence, which were used to study the expression of WOX gene in the developmental process of leaves and stem nodes.

Abiotic stress treatments used 150 mM/L NaCl to simulate salt stress, 0.3 mol/L NaHCO$_3$ to simulate mixed salt and base stress, 20% (*w*/*v*) PEG 6000 to simulate drought stress, and 150 µmol/L CdCl$_2$ to simulate heavy metal stress, and 100 mL of the solution was poured into a culture bowl daily. The roots and the third and fourth functional leaves were taken at 0, 6, 12, 24, 48, and 72 h of treatment, snap−frozen in liquid nitrogen, and stored at −80 °C for RNA extraction and gene expression analysis. To ensure the reliability of the results, the three plants were mixed as biological replicates.

### 2.6. RNA Extraction and SQ−PCR Analysis

Total RNA from the above plant samples was extracted using an RNA Extraction Kit (RP3302, BIOTEKE CORPORATION, Beijing, China) and then, single−stranded cDNA was synthesized using the ReverTra Ace qPCR RT Kit (FSQ−301, TOYOBO (SHANGHAI) BIOTECH CO. LTD., Shanghai, China) and diluted 10−fold as a PCR template. The 20 µL reaction system consisted of 10 µL of 2 × EsTaq Master Mix, 6 µL of double distilled water, 2 µL of template, and 1 µL each of upstream and downstream primers (Table S2). The reaction conditions for SQ−PCR were as follows: initial denaturation at 94 °C for 2 min, followed by 40 cycles of denaturation at 94 °C for 30 s and annealing at 58 °C for 30 s, then a final extension at 72 °C for 10 s. *PsnACT* was used as an internal reference to adjust the loading volume to reach the same concentration of the product, and the corresponding amount of PCR product was taken for 0.8% agarose gel electrophoresis. We performed manual lane selection and standard volume input for the obtained gel images using GIS ID analysis software, with units set to ng, input method to total, and a standard total set. The obtained data were normalized and processed with the following equation:

$$\text{Xnorm} = \frac{X - Xmin}{Xmax - Xmin} \tag{1}$$

*X* and Xnorm are the values before and after normalization, and *Xmin* and *Xmax* are the minimum and maximum values in the sample data, respectively. The obtained data were visualized using Heml software. Three technical replications were performed to ensure the reliability of the results.

To further determine the accuracy of the SQ−PCR results, we randomly selected different genes under different treatments and validated them using Quantitative Real−time PCR (qRT−PCR). Real−time PCR was performed using Power Green qRCR Mix reagent (R0202−02, Dongsheng Biotech Co., Ltd., Beijing, China) based on the SYBR Green fluorescence program. The PCR cycling protocol consisted of initial denaturation at 94 °C for 3 min, followed by 45 cycles of 94 °C for 5 s, 55 °C for 5 s, and 72 °C for 10 s. After the last cycle, a melting curve analysis was performed over a temperature range of 55–94 °C in increments of 1 °C to verify the reaction specificity. The *PsnACT* gene was used as an internal reference gene, and relative expression was measured using the $2^{-\Delta\Delta Ct}$ method. The results were visualized using GraphPad Prism software.

## 3. Results

### 3.1. Cloning, Identification, and Physicochemical Characterization of PsnWOX Sequences

PCR amplification was performed using cDNAs from different tissues of *Populus* × *xiaohei* as templates using the specific primers listed in Table S1, and PCR amplification and sequencing were performed after ligating pMD−18T. The results showed that the lengths of the eight *PsnWOX* genes were 1391, 683, 673, 657, 1124, 852, 682, and 832 bp (Figure 1), and they were highly homologous with the WOX gene of *P. trichocarpa* (Figure S1). Therefore, the above sequences were identified as the ORFs of the *PsnWOX* genes and were named *PsnWOX1a*, *PsnWOX4a*, *PsnWOX4b*, *PsnWOX5b*, *PsnWOX9*, *PsnWOX13a*, *PsnWOX13b*, and *PsnWOX13c*, using the naming strategy of Zhang et al. [1].

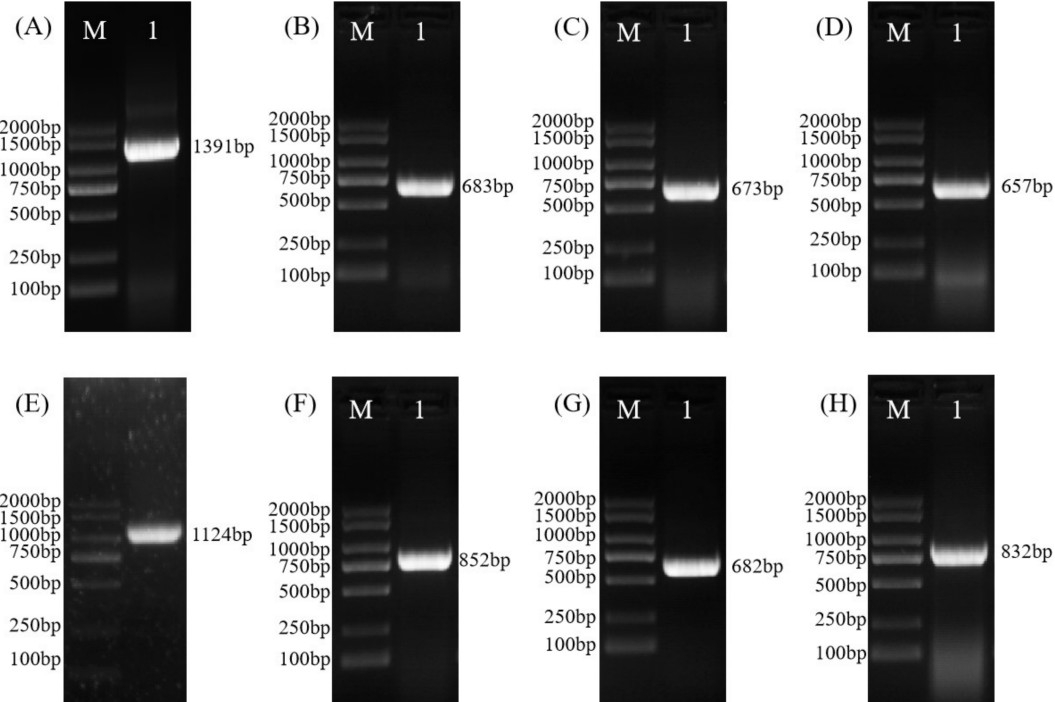

**Figure 1.** The products of PCR amplification of *PsnWOX* genes. M stands for DL 2000 Marker; number 1 stands for the PCR product. (**A**) *PsnWOX1a*; (**B**) *PsnWOX4a*; (**C**) *PsnWOX4b*; (**D**) *PsnWOX5b*; (**E**) *PsnWOX9*; (**F**) *PsnWOX13a*; (**G**) *PsnWOX13b*; (**H**) *PsnWOX13c*.

Physicochemical property analysis showed that the corresponding *PsnWOX* proteins' (*PsnWOXs*) length and molecular weight ranges were 213–390 amino acids and 24506.56–44093.94 Da, and the isoelectric point range was 5.33–9.30. All of the proteins were hydrophilic proteins. Subcellular localization analysis demonstrated that all genes, except *PsnWOX5b*, were located in the nucleus (Table 1).

**Table 1.** Features of the eight WOX genes from *Populus* × *xiaohei*.

| Gene | Chromosome | Protein Length (Amino Acids) | Molecular Weight (Da) | PI | Aliphatic Index | GRAVY | Subcellular Localization |
|---|---|---|---|---|---|---|---|
| *PsnWOX1a* | Chr12 | 388 | 44093.94 | 5.78 | 61.42 | −0.768 | Nucleus |
| *PsnWOX4a* | Chr02 | 213 | 24506.56 | 9.30 | 58.64 | −1.010 | Nucleus |
| *PsnWOX4b* | Chr14 | 213 | 24520.59 | 9.30 | 58.64 | −1.010 | Nucleus |
| *PsnWOX5b* | Chr10 | 219 | 25237.66 | 8.92 | 65.48 | −0.546 | Cytoblast |
| *PsnWOX9* | Chr04 | 390 | 43080.37 | 9.10 | 61.46 | −0.559 | Nucleus |
| *PsnWOX13a* | Chr05 | 248 | 27988.99 | 5.51 | 66.85 | −0.887 | Nucleus |
| *PsnWOX13b* | Chr05 | 216 | 24604.80 | 5.73 | 67.27 | −0.763 | Nucleus |
| *PsnWOX13c* | Chr02 | 215 | 24521.59 | 5.33 | 59.91 | −0.775 | Nucleus |

### 3.2. Multiple Sequence Alignment and Phylogenetic Analysis of PsnWOX Proteins

We analyzed a multiple alignment analysis of the 23 WOX genes using BioEdit software. The results are shown in Figure 2, which shows that the HB structural domain was highly conserved and consisted of 60 amino acid residues forming a helix−loop−helix−turn−helix structure [12]. *AtWUS* consists of 61 amino acid residues due to the insertion of a conserved Y residue after the 17th amino acid [1]. Eleven conserved proteins were reported by Zhang et al. after analysis of the homeodomains of *Sorghum bicolor*, *Z. mays*, *O. sativa*, *A. thaliana*, and *P. trichocarpa*, including Q and L in helix1 and V, W, F, N, and R in helix3, and these conserved proteins were also present in *Populus* × *xiaohei*. The results suggest that these amino acid residues are critical for the function of the *PsnWOX* genes.

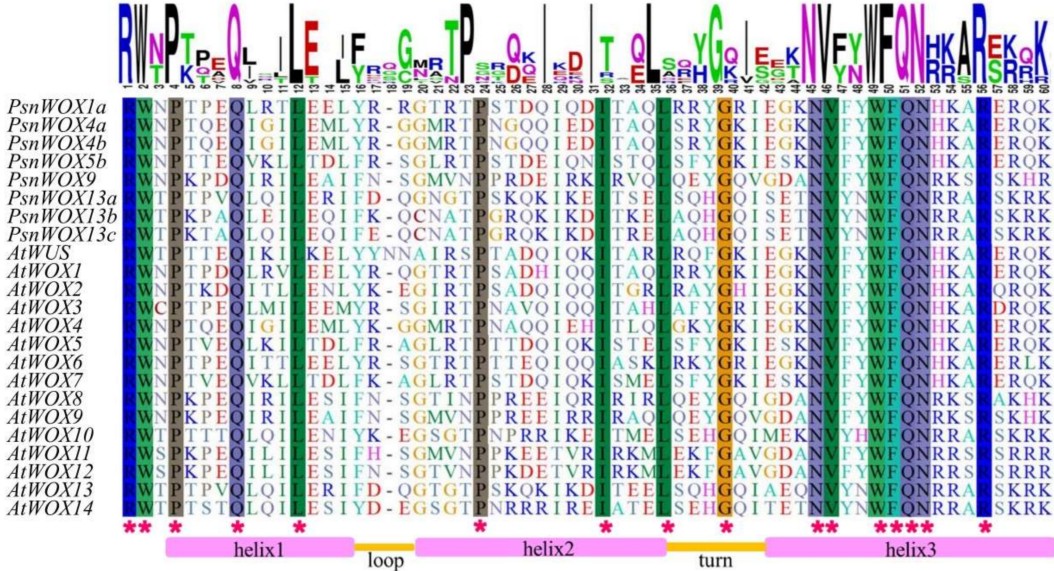

**Figure 2.** Multiple sequence alignment was performed on WOX proteins from *Populus* × *xiaohei* and *A. thaliana*. Sequence logo highlight conserved positions in the sequence comparison, and fully conserved residues are marked with colored underlines and red asterisks (*).

To explore the evolutionary relationships of WOX genes between *Populus* × *xiaohei* and other species, an NJ tree was constructed for 63 WOX proteins from *A. thaliana* (At: 15), *Eucalyptus grandis* (Eg: 9), *O. sativa* (Os: 13), *P. trichocarpa* (Pt: 18), and *Populus* × *xiaohei* (Psn: 8). As shown in Figure 3, the 63 WOX genes divided into three clades: ancient, intermediate, and modern. *PsnWOX1a*, *PsnWOX4a*, *PsnWOX4b*, and *PsnWOX5b* belonged to the modern clade; *PsnWOX9* belonged to the intermediate clade; *PsnWOX13a*, *PsnWOX13b*, and *PsnWOX13c* belonged to the ancient clade, consistent with previous findings [11]. The phenomenon of gene amplification in higher plants resulted in a much lower number

of WOX genes in intermediate and ancient clades than in modern clades. Further analysis showed that the WOX genes of *Populus* × *xiaohei* and *P. trichocarpa* were highly homologous at 99% (Figure S1).

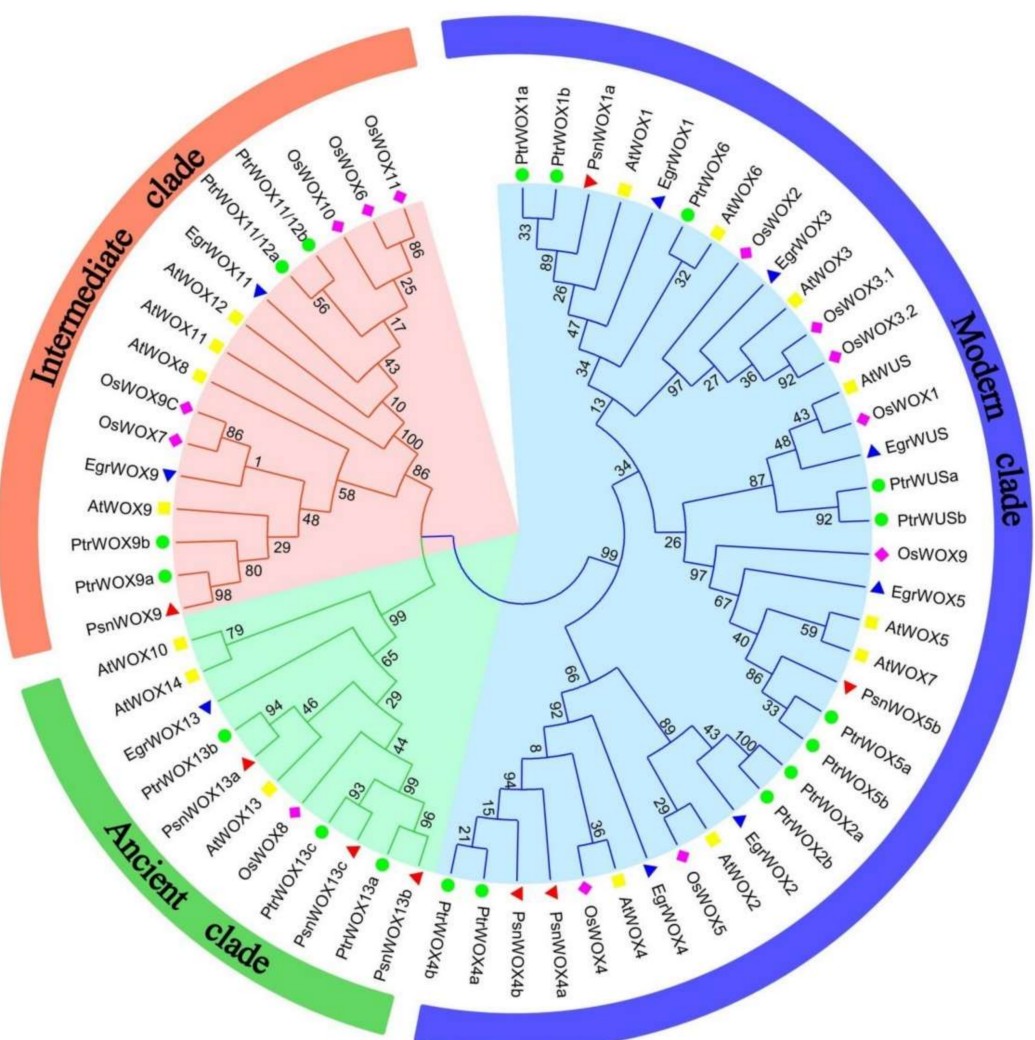

**Figure 3.** Phylogenetic relationships of 63 WOX genes. The WOX genes from *A. thaliana* (At: 15), *E. grandis* (Eg: 9), *O. sativa* (Os: 13), *P. trichocarpa* (Pt: 18), and *Populus* × *xiaohei* (Psn: 8) are shown as yellow squares, blue triangles, purple diamonds, green circles, and red triangles, respectively. Bootstrap values are shown near the nodes.

### 3.3. Exon/Intron Structure and Conserved Motifs Analysis

To further examine the structural diversity of the WOX genes in *Populus* × *xiaohei*, an exon−intron map was constructed using the GSDS 2.0 online site. As shown in Figure 4, the number of introns contained in genes within the same clade was conserved, while the number and position of introns differed in different clades. Two intron insertion sites are highly conserved in the ancient clade, indicating that the *PsnWOX13a*, *PsnWOX13b*, and *PsnWOX13c* genes have very high homology; in the modern clade, the genes *PsnWOX4a* and *PsnWOX4b* have completely conserved intron patterns, and both contain two intron insertion sites. The above results indicate that the genes have undergone replication events during the evolutionary process in the evolution of the gene family. *PsnWOX1a* has an extra intron insertion site that is presumed to be an important locus driving its functional differentiation. Further analysis revealed that the WOX genes of

*Populus* × *xiaohei* and *A. thaliana* are directly homologous with a highly similar organization of exons and introns.

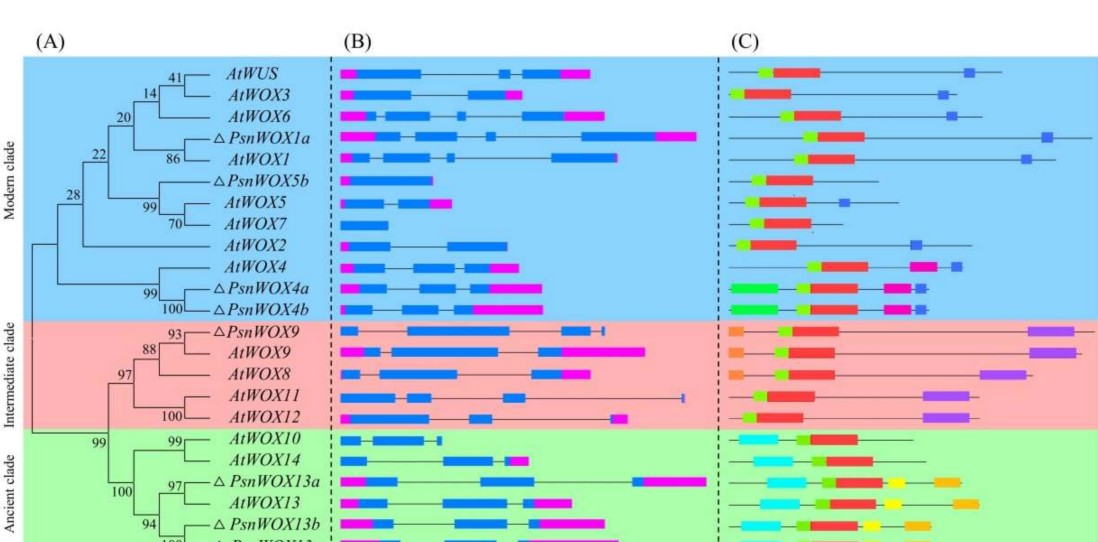

**Figure 4.** Gene structure and conserved motifs of WOX genes in *A. thaliana* and *Populus* × *xiaohei*. (**A**) The NJ tree was constructed from a total of 23 WOX genes from *Populus* × *xiaohei* and *A. thaliana*. The different background colors represent the three clades that are supported by high bootstrap values. (**B**) Gene structures were generated using the GSDS 2.0. (**C**) Conserved motifs were predicted using the MEME tool.

The conserved motifs in *PsnWOXs* were analyzed using MEME analysis software, and a total of 10 conserved motifs were identified and named as motifs 1–10 (Figure 4). The results showed that the *PsnWOX* genes all contained a homeodomain consisting of 60 amino acid residues (motif 1). Further analysis revealed that the subfamilies have similar conserved motif composition, and some motifs are unique to different clades. Motif 2 is present only at the N−terminus in the ancient clade; motif 6 is present only at the C−terminus in the ancient clade, and motif 4 and motif 5 are specific to the WOX4 subfamily of the modern clade. According to Kamiya et al., the rice WUS proteins have an acidic structural domain, a WUS−box, and an EAR−like structural domain [12]. Of these, the acidic domain is a potential transcriptional activation domain in eukaryotes, while the WUS−box motif is only present in members of the WUS clade and is located at the carboxyl terminal of the homologous structural domain, i.e., motif 3. The above results indicate that these motifs are shared by WOX proteins in their evolutionary clade and that they are essential for genes to perform a wide variety of functions. Detailed information is shown in Table S3.

### 3.4. Chromosomal Location, Duplication Analysis and Ka/Ks Calculation

To determine the distribution of the *PsnWOX* genes, we mapped their positions on the poplar chromosomes and the results are shown in Figure 5. The 8 *PsnWOX* genes were distributed on six chromosomes: two on each of chrom 02 and 05, and one on each of chrom 04, 10, 12, and 14.

Genome−wide analysis of *Populus* × *xiaohei* showed that seven fragment duplication events were identified among the 12 *PsnWOX* genes on duplicated fragments (Figure 6), including three pairs of *PsnWOX4a* and *PsnWOX4b*, *PsnWOX13a* with *PsnWOX13c*, and *PsnWOX13b* with *PsnWOX13a*, among the eight genes we cloned out. These results suggest that some *PsnWOX* genes may arise from gene duplication events and suggest that these duplication events are the main drivers of *PsnWOX* gene family amplification.

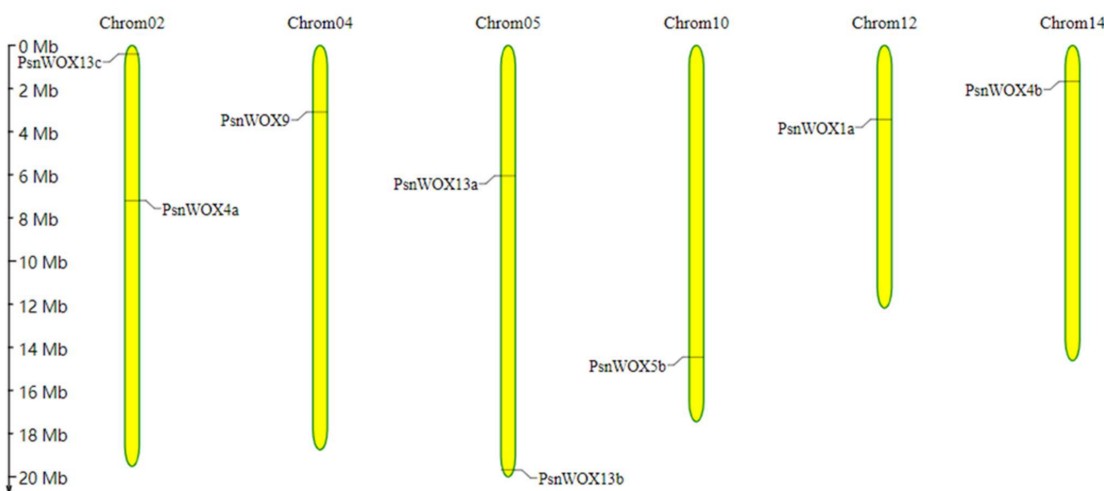

**Figure 5.** Chromosomal distribution of *PsnWOX* genes. Yellow strips represent chromosomes. Chromosome numbers are shown above the bar chart.

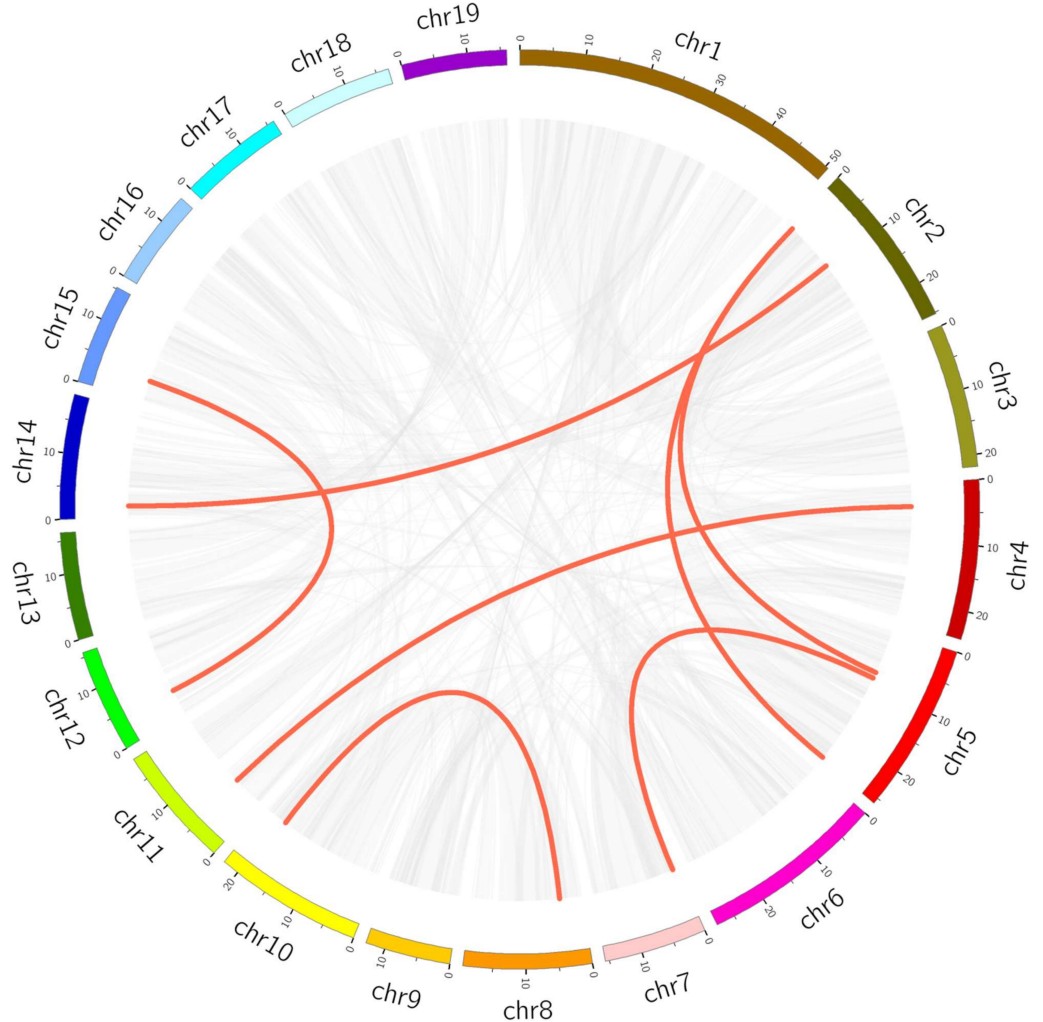

**Figure 6.** Schematic representations of segmental duplications of *PsnWOX* genes. The seven putative segmental duplication pairs are linked by the colored lines.

To further understand the gene duplication mechanism of the WOX gene family in Populus × xiaohei, a comparative map with the dicotyledonous plants A. thaliana and Glycine max and the monocotyledonous plant O. sativa was constructed (Figure 7). The number of homologs between Populus × xiaohei and A. thaliana and G. max were 16 and 37, respectively, much higher than the 2 in rice, indicating a strong direct homology between the Populus × xiaohei WOXs and the dicotyledons members, which showed a high degree of evolutionary divergence compared with the monocotyledons. WOX genes containing more co−connection gene pairs may have played an important role in the evolution of the Populus × xiaohei WOX gene family, such as PsnWOX5, which contains three co−connection gene pairs. The ratio of Ka/Ks can effectively improve the understanding of the evolutionary constraints of the WOX gene family, and the calculated results are shown in Table S4. Positive or Darwinian selection requires sequences with Ka/Ks > 1; neutral drift requires Ka/Ks = 1; and purifying selection requires Ka/Ks < 1 [43]. It follows that duplicated gene pairs are mainly under purifying selection conditions (Ka/Ks < 1.0).

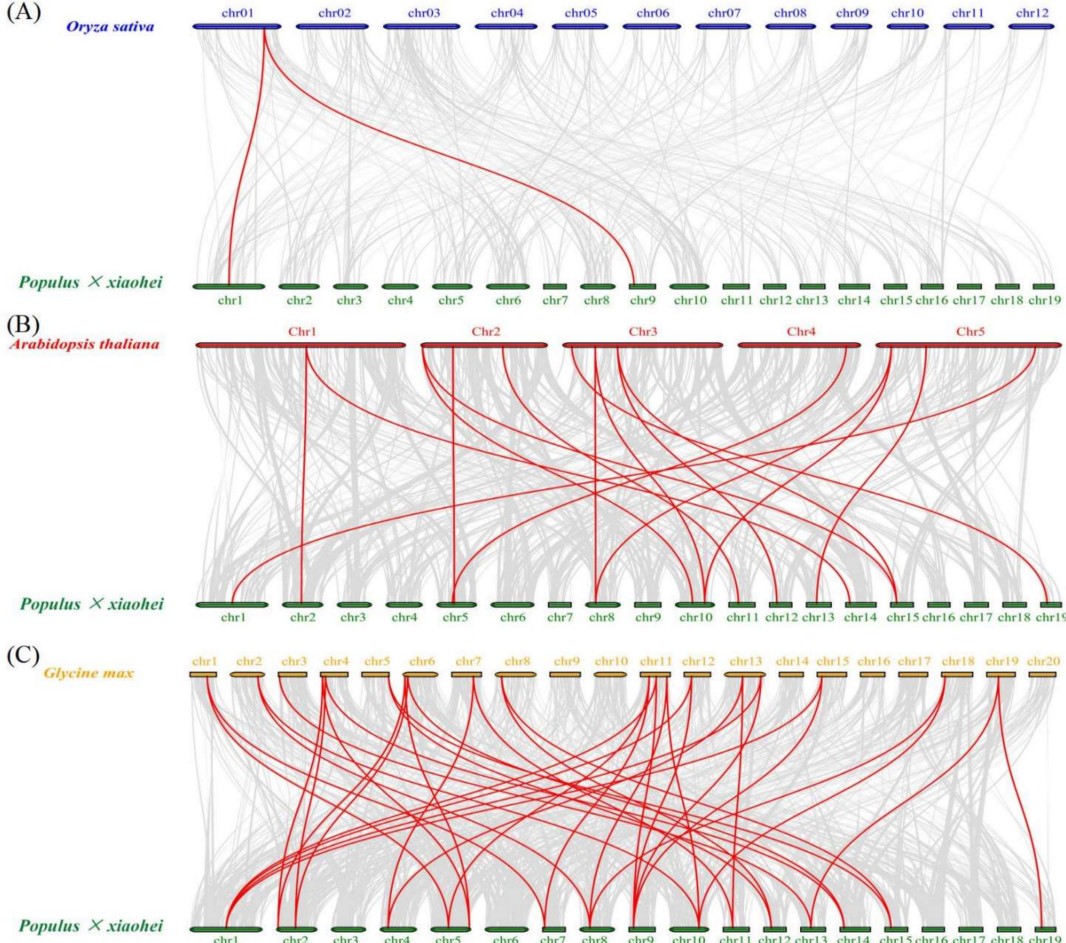

**Figure 7.** Synteny analysis of WOX genes between *Populus* × *xiaohei* and other plants. (**A**) Monocotyledonous plant *O. sativa*. (**B**) Dicotyledonous plants *A. thaliana*. (**C**) Dicotyledonous plants *G. max.* Grey lines in background indicate collinear blocks within *Populus* × *xiaohei* and other plant genomes; red lines indicate syntenic WOX gene pairs.

### 3.5. Tissue−Specific Expression Pattern of PsnWOX Genes

The expression characteristics of WOX genes vary greatly in different species. For example, Liu et al. found that many WOX genes in *Gossypium hirsutum* have different expression patterns when compared with their homologs in *A. thaliana* [44]. To reveal the functions of this family of genes in woody plant growth and development, we ana-

lyzed the expression characteristics of eight *PsnWOXs* in different tissue sites by SQ−PCR. The results are shown in Figure 8 and Figure S2.

**Figure 8.** Tissue−specific expression pattern of *PsnWOX* genes. (**A**) Different tissue sites; (**B**) Stem nodes throughout the development period; (**C**) Leaves throughout development.

The expression characteristics of *PsnWOX* genes in terminal buds, leaves, xylem, phloem, and roots were analyzed, and the results as shown in Figure 8 showed that there were significant differences in the expression of *PsnWOX* genes in different tissues. The expression patterns of the three members of the *WOX13* subfamily were generally similar, showing the highest expression in xylem and phloem, followed by roots, and the lowest expression in leaves, indicating that these genes play important roles in plant development. *PsnWOX1a* had the highest expression in leaves and terminal buds; *PsnWOX4* was mainly expressed in xylem, and *PsnWOX5* had a lower expression level in all plant parts.

Most of the *PsnWOXs* were expressed in both the xylem and the phloem of mature stems, and the expression of these genes was further analyzed during stem development. *PsnWOX1a* consistently had high expression levels at the early and late stages of stem node development. The expression patterns of WOX genes within the same subfamily remained relatively consistent. *PsnWOX13b* and *PsnWOX13c* showed a continuous increase in expression to a maximum at the early stage of node development, followed by a slight downregulation, and then remained relatively stable, suggesting that the two genes play important regulatory roles in the early stages of node development. *PsnWOX4a* and *PsnWOX4b* showed continuous decreases in expression throughout node development. The above results suggest that *PsnWOXs* have different functions during the development of stem nodes.

To analyze the expression of WOX genes at different stages of leaf development, the entire course of development from young to senescent leaves was selected, and the first leaf was used as a control. *PsnWOX1a* was consistently expressed at a high level during early and late leaf development; *PsnWOX5b, PsnWOX13b,* and *PsnWOX13c* were expressed at lower levels at the beginning of leaf development and then gradually increased, indicating that they performed their corresponding functions during late leaf development. *PsnWOX13a* was significantly less expressed in mature leaves than *PsnWOX13b* and *PsnWOX13c*, and there were slight differences in their expression patterns, suggesting that the three genes derived from replication events may have been altered with respect to their functions in the leaves.

*3.6. Expression Patterns of PsnWOX Genes under Different Stress Treatments*

WOX genes are involved in drought and salt damage response in *Gossypium hirsutum* and *P. alba* × *P. glandulosa* [45,46]. To investigate the effect of abiotic stress on the expression of the WOX genes in *Populus* × *xiaohei*, we counted its expression in roots and leaves under four stresses, CdCl$_2$, NaCl, NaHCO$_3$, and PEG. The results are shown in Figure 9 and Figure S2.

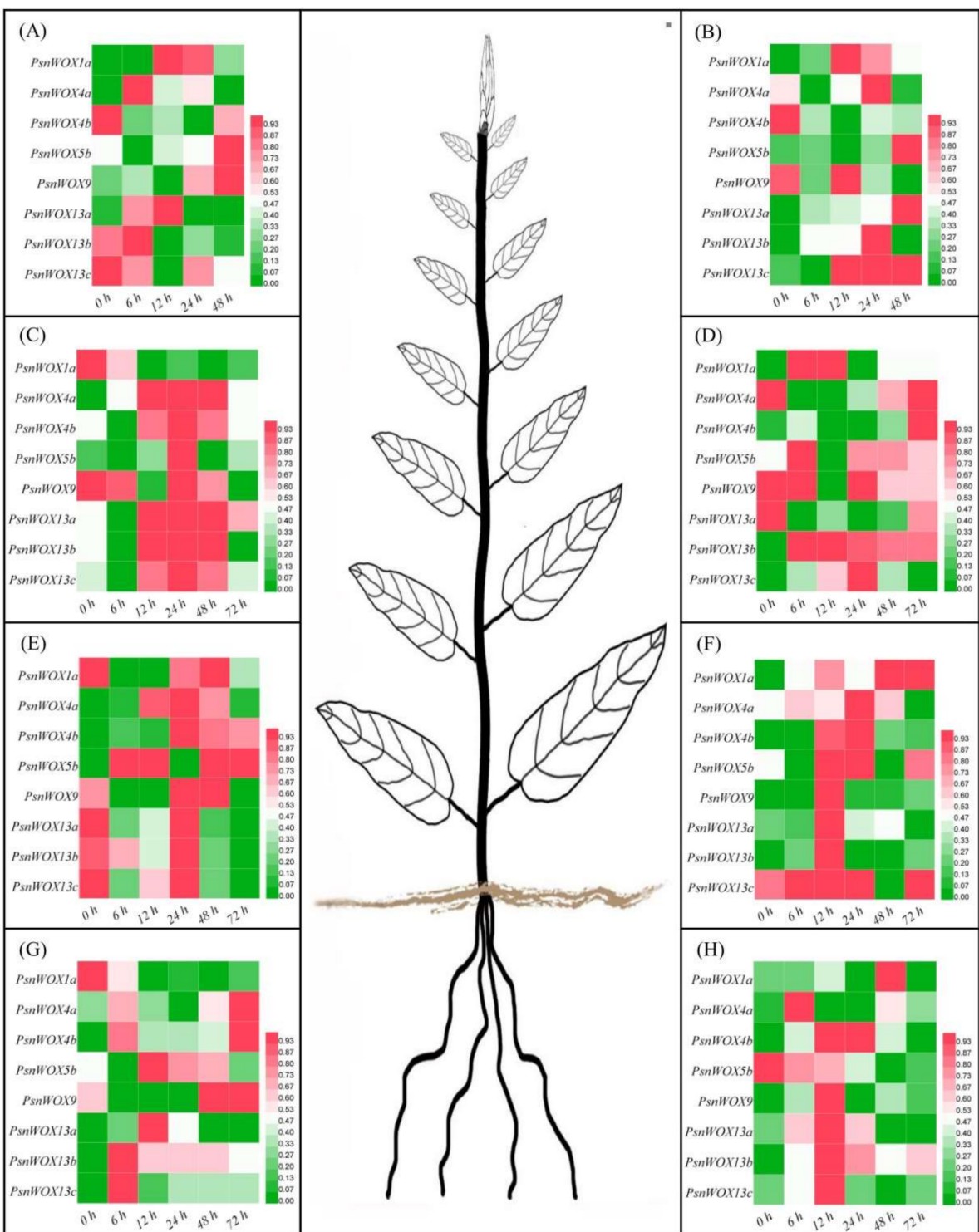

**Figure 9.** *PsnWOX* gene expression patterns under different stress treatments. The relative expression levels of WOX genes in two different tissues (roots and leaves) of *Populus × xiaohei* seedlings treated with CdCl$_2$, NaCl, NaHCO$_3$, or PEG were analyzed by SQ−PCR. The data represent the mean ± SE of three independent replicates. (**A**) Expression of CdCl$_2$ treatment in roots; (**B**) Expression of CdCl$_2$ treatment in leaves; (**C**) Expression of NaCl treatment in roots; (**D**) Expression of NaCl treatment in leaves; (**E**) Expression of NaHCO$_3$ treatment in roots; (**F**) Expression of NaHCO$_3$ treatment in leaves; (**G**) Expression of PEG treatment in roots; (**H**) Expression of PEG treatment in leaves.

The expression of *PsnWOX* genes under CdCl$_2$ treatment is shown in Figure 9. *PsnWOX13a* and *PsnWOX13b* showed a positive regulation pattern in the early stage of treatment, after which the expression decreased sharply; the expression of *PsnWOX13c* based on replication events continued to be downregulated throughout the stress process, suggesting that CdCl$_2$ may have suppressed the expression of *PsnWOX13c* through the corresponding signal transduction pathway. *PsnWOX1a* in modern clades showed mainly upregulated expression during the stress treatment, with the gene expression reaching its highest value at 12 h, after which the expression was downregulated and stabilized. The expression of genes *PsnWOX5b*, *PsnWOX13a*, and *PsnWOX13c* in leaves was consistently elevated within 48 h of the stress treatment.

The response pattern of *PsnWOX* genes to NaCl stress is shown in Figure 9. NaCl treatment inhibited the expression of *PsnWOX1a* in the roots, and its expression was downregulated with increasing treatment time. Three genes in the *PsnWOX13* subfamily showed similar expression patterns in roots, with a transient downregulation of expression at 6 h, followed by a rapid increase and leveling off, and a slight downregulation of expression after 48 h, indicating that these genes responded to NaCl stress. The expression of *PsnWOX4a*, *PsnWOX4b*, *PsnWOX5b*, and *PsnWOX9* all increased to their highest levels at 24 h in roots, and *PsnWOX5b* increased significantly at 6 h in leaves.

The expression patterns of the WOX genes in the subfamily remained consistent in the NaHCO$_3$−treated roots. Expression levels of all three genes in the WOX13 subfamily increased to the highest values at 24 h of treatment, and then their expression gradually decreased to below initial levels. The expression of two genes in the WOX4 subfamily increased at 24 h and decreased slightly with the increase of stress treatment time but was still above the initial level. Although *PsnWOX1a* and *PsnWOX9* were in different evolutionary clades, their expression patterns were similar throughout the treatment, with their expression decreasing at the beginning of the stress treatment, being rapidly upregulated at 24 h and continuing until 48 h, after which their expression again decreased. In leaves, the expression levels of *PsnWOX9*, *PsnWOX13a*, *PsnWOX13b*, and *PsnWOX13c* all peaked at 12 h after stress treatment and then decreased slightly and leveled off.

In PEG−treated roots, drought stress inhibited the expression of *PsnWOX1a*, and its relative expression in roots was downregulated with increasing treatment time, while expression was induced in the remaining genes. The genes *PsnWOX13b* and *PsnWOX13c* in the ancient clade showed a significant upregulation at 6 h of drought treatment, indicating that *Populus* × *xiaohei* responds to drought stress by regulating the expression level of the *PsnWOX13b* gene; the gene *PsnWOX5b* in the modern clade had higher expression, increasing rapidly at 12 h and lasting until 48 h, after which the expression decreased; in the leaves, *PsnWOX4a*, *PsnWOX4b*, *PsnWOX9*, *PsnWOX13a*, *PsnWOX13b*, and *PsnWOX13c* showed mainly upregulated expression in the early stage of stress (12 h or 24 h), while the expression decreased slightly afterwards.

SQ−PCR is a common method for gene expression analysis, and to determine the correctness of the trend of the SQ−PCR results, they were compared with the qRT−PCR results (Figure 10). The results showed the same overall trend, and the qRT−PCR results further proved the accurate line of SQ−PCR results.

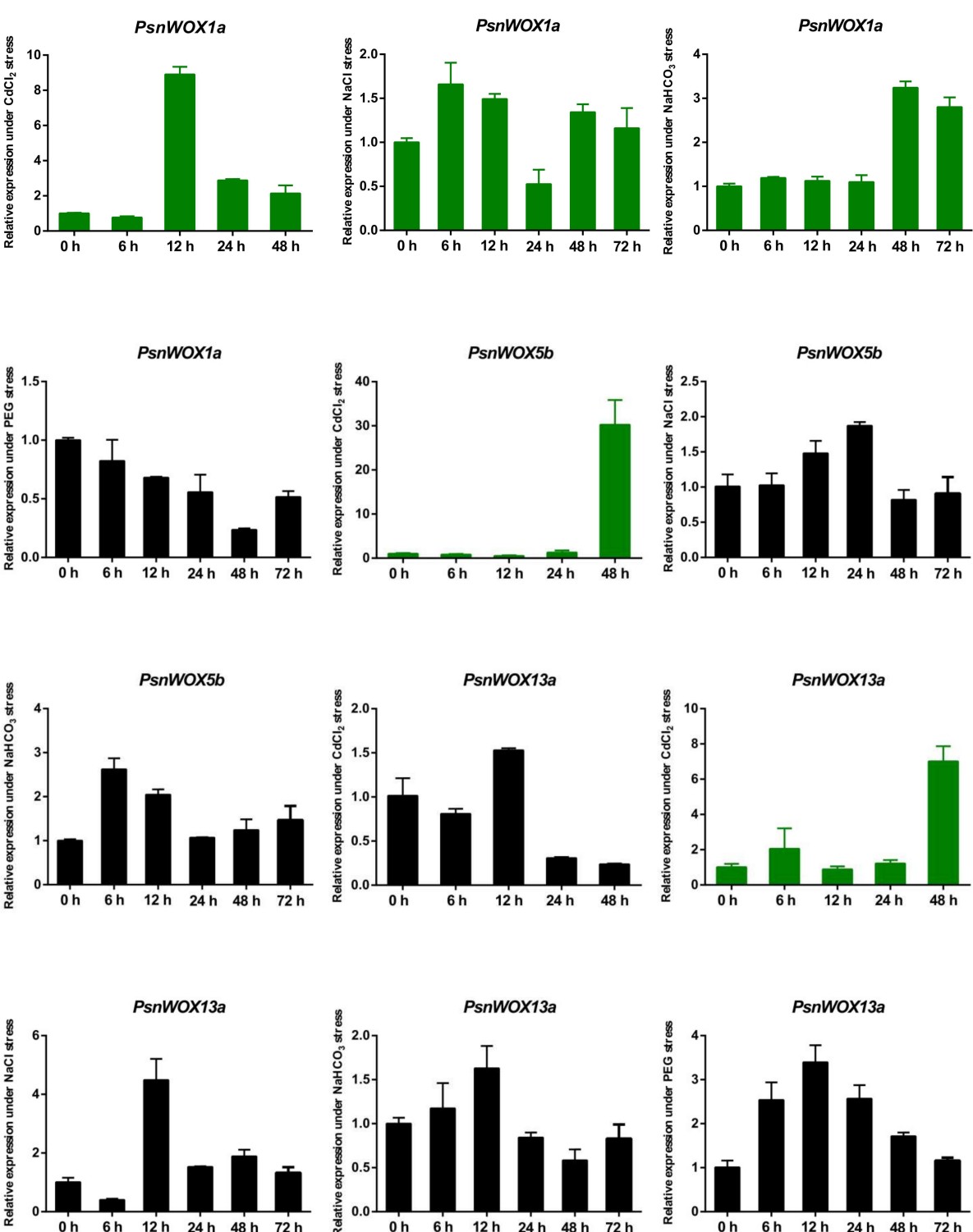

**Figure 10.** qRT−PCR analysis of some *PsnWOXs* expression to verify SQ−PCR results. Transcript levels of *PsnWOXs* were calculated using the $2^{-\Delta\Delta Ct}$ method. Green columns represent the expression in the leaf, black column represents the expression in the root.

## 4. Discussion

WOX genes have been identified and studied in many plants [47–50]. Phylogenetic analysis showed that the 63 WOX genes were divided into three clades, ancient, intermediate, and modern (Figure 3), and this result was consistent with the results in

*Brassica napus* [51], *Panax ginseng* [52], and *Rosa rugosa*. The phenomenon of gene amplification in higher plants resulted in a much lower number of WOX genes in intermediate and ancient clades than in modern clades [53]. A helix–loop–helix–turn–helix structure in the HB domain allows it to recognize sequence−specific targets. Furthermore, the domain is conserved in *Z. mays* [7], *Petunia hybrid* [54], and *Vitis vinifera*, thus maintaining its functional integrity. Compared with the conserved site of *PsnWOXs*, *AtWUS* has an extra Y residue in the loop structure [1], as do *PtrWUS*, *OsWUS*, and *ZmWUS*. The specific function of the Y residue has not been reported in the literature, and subsequent molecular biology experiments such as mutation of the Y residue can be performed to elaborate its specific biological function.

Analysis of the structural features of the *PsnWOX* genes revealed that the insertion sites of introns in the same clade were similar, further supporting the delineation of subclades and functional conservation. Outside the HB structural domain, the majority of conserved motifs are unique to each clade, implying their critical importance in the unique functions of the proteins. In addition, the WUS box motif is only present in members of the WUS clade and is located at the carboxyl terminus of the homologous structural domain [12].

The above results suggest that specific motifs are present in certain clade and subclade proteins, and presumably they are significant for maintaining functional conservation. In general, gene families expand by tandem duplication and segmental duplication. A genomewide analysis indicated that *PsnWOX* genes were subjected to whole−genome replications and tandem repeat events [55]. Seven tandem repeat events between twelve genes may have contributed to the evolution of the WOX gene family. The selection stress analysis indicated that continuous purification selection (Ka/Ks < 1) has played a key role in determining the number of WOX genes in *Populus × xiaohei*.

Analysis of WOX gene expression characteristics revealed that gene expression patterns and expression levels were generally similar for the same clade, but there were differences in the expression of different genes in different tissue sites and under various abiotic stresses. WOX proteins create a proximal differentiation gradient by repressing leaf proximal differentiation and promoting leaf distal differentiation, a regulatory pattern that allows for sustained leaf growth and allows the leaf to achieve its final shape [56–58]. The *RcWOX1* gene was expressed only in the roots, and overexpression plants exhibited a significant increase in the number of lateral roots [59]. The homologous gene *PsnWOX1* is highly expressed in leaves, leading to the speculation that it also functions in regulating lateral leaf growth. *AtWOX4* is mainly expressed in the vascular cambium and procambium, maintaining the number and status of vascular cambium stem cells, and the *OsWOX4* gene is also highly expressed in the inflorescence and floral meristem [60].

It has been shown that the lateral meristem development involving the *WOX4* gene is dependent on certain growth hormone levels [61]. In this study, we found that *PsnWOX4* was highly expressed in the xylem and that this may depend on certain growth hormone levels in the maintenance of vascular cambium stem cells under salt and drought stress. *WOX5* was expressed in the apical meristematic region (RAM) of *A. thaliana* roots and was able to promote stem cell proliferation, with overexpression lines exhibiting short primary stems of seedlings or even failure to shoot [62]. *TaWOX5* was also abundantly expressed in wheat roots; in poplar, *WOX5a* was mainly expressed in adventitious root tips and lateral root tips, and its overexpression led to an increase in AR numbers and a decrease in AR lengths and leaf numbers.

*PsnWOX5b* was detected in all tissue sites, presumably not only maintaining stem cell stability in root apical meristematic tissue but also responding to abiotic stress by regulating its expression level in roots. *WOX8* and *WOX9* are expressed during embryo development in *A. thaliana* [63], and their roles in regulating shoot structure are conserved across species. *PsnWOX9* showed an upregulation trend at the late stage of stem development, with its expression peaking at the 11th stem node, presumably acting at the stem node maturation stage and co−involved in the regulation of shoot structure. *AtWOX13* has high expression levels in inflorescence, flower buds, lateral roots, and root tips [64,65];

*PtrWOX13* is expressed in several tissues including the xylem and the phloem, and it affects the nutritional growth and secondary tissue formation in *P. trichocarpa*. The *PsnWOX13* subfamily genes, which are homologous in *Populus × xiaohei*, showed different levels of upregulated expression in the xylem, the phloem, and roots and may be involved in plant secondary growth and root development.

Our data suggest that a few *PsnWOX* genes have different expression patterns from their homologs in *A. thaliana* and differ somewhat from the expression in *P. trichocarpa* (Figure S3). *PsnWOX5* was expressed at very low levels in roots and differed from the expression profile of *WOX5* in *Populus tomentosa* [44]. We also noted that duplicated genes exhibited different expression levels, and this variability was present in species such as cotton [66]. *PsnWOX4a* was highly expressed in the phloem, whereas *PsnWOX4b* was difficult to detect at the same site; *PsnWOX13b* and *PsnWOX13c* were significantly more highly expressed in the terminal buds than *PsnWOX13a*. These results suggest that some WOX genes may have acquired different functions during evolution and that some differences exist between species [67,68].

After plants are subjected to drought and salt stress, a series of physiological and biochemical changes will occur in vivo, including re−establishing cellular ion homeostasis, repairing organismal injury, and coordinating growth regulation. In this process, signal transduction factors and transcription factors are important, as they can receive signals and transmit them to downstream transcription factors to regulate gene expression. For example, most of the constitutive *BpWOX* genes are tolerant to drought, salt, cold, and cadmium ($CdCl_2$) [28]. Cadmium stress alters the expression of paralogous homologous WOX genes through cytokine accumulation, thereby affecting aboveground form and root size and shape; WUS/WOX5−related genes and cytokine signaling play key roles in the regulatory network of SAM and RAM maintenance and activity, and cadmium induces their misexpression [69–71].

In $CdCl_2$−treated roots, *PsnWOX13a* and *PsnWOX13b* exhibited a positive regulatory pattern in the pretreatment period, presumably due to having a regulatory role in the corresponding abiotic stress process by responding to cytokines, whereas *PsnWOX13c*, which is based on replication events, differed in function, and its expression was continuously downregulated throughout the stress process, suggesting that $CdCl_2$ stress may have suppressed its expression through the corresponding signal transduction pathway. In rice, the expression of *OsWOX3* and *OsWOX5* could be rapidly increased 3−fold to 8−fold after 1 h treatment with NaCl, while other WOX genes did not respond to NaCl stress [72].

The expression of *PsnWOX5b* also increased rapidly in the early stage, but other genes differed from rice, showing that the expression of all genes, except *PsnWOX1a*, in the roots was strongly induced at 24 h, indicating that these genes could respond rapidly during NaCl stress. In NaHCO$_3$−treated roots, the expression of all *PsnWOX* genes was induced, indicating that *PsnWOX* genes are involved in salt stress resistance. In rice, eight WOX genes (*OsWUS*, *OsWOX3*, *OsWOX4*, *OsWOX5*, *OsWOX9B*, *OsWOX11*, *OsWOX12A*, and *OsWOX12B*) were upregulated nearly 4−fold after 3 h of induction due to drought; the *OsWOX13* gene is ABA−responsive and is widely involved in regulating seeds and nutrient organs, and its overexpression results in premature flowering and extensive effects on biological processes following drought and salt stress [73,74]. In tomato, induced stress analysis revealed that *SlWOX13* showed a significant upregulation under 7−d drought treatment conditions [34]; the expression of almost all *PsnWOX* genes was induced in drought stress, and the expression was rapidly upregulated when the treatment time reached 12 h, consistent with the results in rice. *PsnWOX13b* was significantly upregulated at 6 h of drought treatment, suggesting that *Populus × xiaohei* may respond to ABA and regulate various postembryonic developmental processes by regulating its expression level in roots in response to drought stress.

## 5. Conclusions

In this study, eight WOX genes were obtained by cloning, and bioinformatic analysis showed that all contained a conserved homeodomain. *PsnWOX1a*, *PsnWOX4a*, *PsnWOX4b*, and *PsnWOX5b* belonged to the modern clade; *PsnWOX9* belonged to the intermediate clade, and *PsnWOX13a*, *PsnWOX13b*, and *PsnWOX13c* belonged to the ancient clade. Seven segmental−duplicated gene pairs among the *PsnWOX* gene family were mainly under purifying selection conditions. The expression of family members differed in different tissues, with *PsnWOX1a* mainly expressed in leaves, the *PsnWOX4* subfamily mainly concentrated in the xylem and phloem, and the *PsnWOX13* subfamily having high expression in the xylem, phloem, and roots. SQ−PCR showed that WOX genes responded strongly and were differentially expressed under four stress treatments, CdCl$_2$, NaCl, NaHCO$_3$, and PEG. The expression of *PsnWOX4a*, *PsnWOX4b*, *PsnWOX9*, *PsnWOX13a*, *PsnWOX13b*, and *PsnWOX13c* reached maximum levels in roots at 24 h under NaCl and NaHCO$_3$ stress; the amount of expression of the *PsnWOX13* subfamily was significantly increased in the early stages of CdCl$_2$ treatment, and the expression of almost all of the *PsnWOX* genes was induced under drought conditions. This study provides important information concerning the WOX genes of *Populus × xiaohei* and provides a theoretical reference for their functions in abiotic stresses.

**Supplementary Materials:** The following supporting information can be downloaded at: https://www.mdpi.com/article/10.3390/f13010122/s1. Table S1: Primers for *PsnWOX* gene cloning, Table S2: SQ−PCR primers for *PsnWOX* gene, Table S3: Motif sequences of *PsnWOX* proteins, Table S4: Information on the Segmental duplications analysis of the *PsnWOX* genes. Figure S1: Phylogenetic tree of *Populus × xiaohei* and *P. trichocarpa*, Figure S2: SQ−PCR electrophoresis profiles of the WOX gene of *Populus × xiaohei* in tissue sites and under abiotic stress, Figure S3: Tissue−specific expression profiles of *PsnWOX* genes.

**Author Contributions:** Conceptualization, H.L. and Y.L. (Yue Li); methodology, H.L.; software, Y.L. (Yue Li), F.L. and B.W.; validation, Y.L. (Yue Li) and H.L.; formal analysis, Y.L. (Yue Li); investigation, Y.L. (Yue Li), C.J., Y.L. (Yuting Liu), and L.W.; resources, Y.L. (Yue Li); data curation, Y.L. (Yue Li) and H.L.; writing—original draft preparation, Y.L. (Yue Li); writing—review and editing, H.L.; visualization, Y.L. (Yue Li); supervision, H.L.; project administration, H.L.; funding acquisition, H.L., G.L. and J.J. All authors have read and agreed to the published version of the manuscript.

**Funding:** This research was funded by the Major Special Project on Breeding New Varieties of Genetically Modified Organisms, grant number 2018ZX08020002.

**Institutional Review Board Statement:** Not applicable.

**Informed Consent Statement:** Not applicable.

**Data Availability Statement:** Not applicable.

**Conflicts of Interest:** The authors declare no conflict of interest.

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
