# Peer review of "Global Analysis of the WOX Transcription Factor Gene Family in Populus × xiaohei T. S. Hwang et Liang Reveals Their Stress−Responsive Patterns"

_forests, doi:10.3390/f13010122_

Round 1

Reviewer 1 Report

Information about WOX genes in woody plants should be added to the Introduction section.

The conditions of the experiment, the results of which are given in section 3.4, should be described in the Materials and Methods (tissue types, developmental stages, etc.).

“T. S. Hwang et Liang” after the first mention in the text, you do not need to write.

L.128. Nutrient bowls are plant growing containers?

L.132-134. It should be indicated which types of abiotic stress were simulated with NaCl, NaHCO3, PEG, and CdCl2.

L.175. "Analysis of the 23 PsnWOXs". In fact, in Fig. 2 indicates 8 PsnWOX and 15 AtWOX. Please correct.

L.179. Eleven or twelve? P, Q, and L in helix1, P and I in helix2, and N, V, W, F, Q, N, and R in helix3 (Fig. 2).

Reviewer 2 Report

In this manuscript author did a global Analysis of the WOX transcription factor gene family in Populus × xiaohei T. S. Hwang et Liang, reveals their stress-responsive patterns. In this study, eight WOX genes were obtained from an endemic Chinese resilient tree species, Populus × xiaohei T. S. Hwang et Liang. Bioinformatic analysis showed that the WOX genes all contained a conserved structural domain consisting of 60 amino acids, with some differences in physicochemical properties. Phylogenetic analysis revealed that WOX members were divided into three evolutionary clades, with four, one, and three members in the ancient, intermediate, and modern evolutionary clades, respectively. The conserved structural domain species as well as the organization and gene structure of WOX genes within the same subfamily, were highly uniform. Semi-quantitative interpretation (SQ-PCR) analysis showed that the WOX gene was differentially expressed in different tissues, and it was hypothesized that the functions performed by different members were diverse. The family members were strongly and differentially expressed under CdCl2, NaCl, NaHCO3, and PEG treatments, suggesting that WOX genes function in various aspects of abiotic stress defense responses. For the betterment, I have a few questions for the authors.

  1. Figure 1A and 1D almost look the same. I want to see a full raw gel picture.
  2. Why do the authors have to do cloning for identifying the populous gene? Does the primer have a full-length sequence covered?
  3. After cloning, what did the author do with that result?
  4. Please validate at least one gene function for stress.
  5. Do collinearity analysis of WOX genes from another gene family along with Populus.
  6. Do Protein interaction network of WOXs in Populus.
  7. What is SQ-PCR? As far as I know, it is semi-quantitative PCR, and I did not see any RT-PCR gel here.
  8. Where is the RT-qPCR result to validate the expression results? You have to do RT-qPCR to validate the expression result
  9. Use Hypothetical figures to explain the results of this study.
  10. The introduction is short. The author should include recent genome-wide studies such as: Genome-Wide Identification, and Characterization of PIN-FORMED(PIN) Gene Family Reveals Role in Developmental and Various Stress Conditions in Triticum aestivum. b. Genome-wide identification and expression pattern analysis of the KCS gene family in barley. c. Genome-wide identification and characterization of abiotic stress-responsive lncRNAs in Capsicum annuum. d. Genome-Wide Identification and Characterization of the Brassinazole-resistant (BZR) Gene Family and Its Expression in the Various Developmental Stage and Stress Conditions in Wheat (Triticum aestivum). e. Genome-wide identification and functional characterization of natural antisense transcripts in Salvia miltiorrhiza. f. Genome-wide identification and expression analysis of the AT-hook Motif Nuclear Localized gene family in soybean.

Round 2

Reviewer 2 Report

I am happy with the author's reply. The authors did add my suggested reference but only in text not in the reference, please correct it. Also, make RT-qPCR result as the main figure. After the author makes changes MS can be accepted for publication.

Author Response

We have added reference based on the reviewers' comments.

We have seen gene expression analysis by SQ-PCR in many literatures, so we also performed related experiments by SQ-PCR.

An, Ma, Du, et al. Preliminary Classification of the ABC Transporter Family in Betula halophila and Expression Patterns in Response to Exogenous Phytohormones and Abiotic Stresses[J]. Forests, 2019, 10(9):722.

Zhou M, Li D, Li Z, et al. Constitutive expression of a miR319 gene alters plant development and enhances salt and drought tolerance in transgenic creeping bentgrass. Plant Physiol. 2013;161(3):1375-1391.

Mukhopadhyay P, Tyagi AK. OsTCP19 influences developmental and abiotic stress signaling by modulating ABI4-mediated pathways [published correction appears in Sci Rep. 2015;5:12381]. Sci Rep. 2015;5:9998. Published 2015 Apr 29.

Zhang X, Dou L, Pang C, et al. Genomic organization, differential expression, and functional analysis of the SPL gene family in Gossypium hirsutum. Mol Genet Genomics. 2015;290(1):115-126.

We currently do not have enough template to perform qRT-PCR experiments, and it would take at least three months to re-perform cuttings, abiotic stress treatments, and sampling of Populus × xiaohei. Therefore, we randomly selected different genes for treatment under different stresses for qRT-PCR to verify the feasibility of SQ-PCR and the results have been put into the text (see Figure 10).